# Screening of *Lactiplantibacillus plantarum* NML21 and Its Maintenance on Postharvest Quality of *Agaricus bisporus* through Anti-Browning and Mitigation of Oxidative Damage

**DOI:** 10.3390/foods13010168

**Published:** 2024-01-03

**Authors:** Chengrui Shi, Xiaoli Yang, Pengjie Wang, Hao Zhang, Qihui Wang, Bo Wang, William Oyom, Weibing Zhang, Pengcheng Wen

**Affiliations:** 1College of Food Science and Engineering, Gansu Agricultural University, Lanzhou 730070, China; scr9809@163.com (C.S.); wqh1030018113@163.com (Q.W.); 2Gansu Institute of Business and Technology Co., Lanzhou 730070, China; yangxl202209@163.com; 3College of Food Science and Nutritional Engineering, China Agricultural University, Beijing 100083, China; wpj1019@cau.edu.cn (P.W.); zhanghaocau@cau.edu.cn (H.Z.); 4Lanzhou Customs Technology Center, Lanzhou 730070, China; wyy080214@163.com; 5Food and Nutritional Sciences Program, North Carolina Agricultural and Technical State University, Greensboro, NC 27411, USA; woyom@aggies.ncat.edu; 6Functional Dairy Products Engineering Lab., Gansu Agricultural University, Lanzhou 730070, China

**Keywords:** *Agaricus bisporus*, lactic acid bacteria, anti-browning, reactive oxygen species, shelf life

## Abstract

Browning and other undesirable effects on *Agaricus bisporus* (*A. bisporus*) during storage seriously affect its commercial value. In this study, a strain, *Lactiplantibacillus plantarum* NML21, that resists browning and delays the deterioration of *A. bisporus* was screened among 72 strains of lactic acid bacteria (LAB), and its preservative effect was analyzed. The results demonstrated that gallic acid, catechin, and protocatechuic acid promoted the growth of NML21, and the strain conversion rates of gallic acid and protocatechuic acid reached 97.16% and 95.85%, respectively. During a 15 d storage of the samples, the NML21 treatment displayed a reduction in the browning index (58.4), weight loss (2.64%), respiration rate (325.45 mg kg^−1^ h^−1^), and firmness (0.65 N). The treatment further inhibited *Pseudomonas* spp. growth and polyphenol oxidase activity, improved the antioxidant capacity, reduced the accumulation of reactive oxygen species, and reduced the malonaldehyde content and cell membrane conductivity. Taken together, the optimized concentrations of NML21 may extend the shelf life of *A. bisporus* for 3–6 d and could be a useful technique for preserving fresh produce.

## 1. Introduction

The button mushroom, *Agaricus bisporus* (*A. bisporus*), accounts for 32% of the total global production of edible mushrooms [1]. However, it lacks protective structures and is highly susceptible to mechanical damage and pathogen infection after harvesting. These results in the autolysis of tissue cells and the destruction of the original cellular compartments, which initiate enzymatic browning from the interaction of phenolic compounds and polyphenol oxidase (PPO) in the cell contents [2]. In addition, browning and other degradation phenomena are exacerbated by the reactive oxygen metabolism of *A. bisporus* itself [3].

The existing preservative techniques for postharvest *A. bisporus* are mainly focused on chemical bacterial inhibitors, modified atmospheric packaging (MAP), and irradiation treatment [4]. However, these methods may raise food safety and consumer health concerns due to chemical residues and the modification of the chemical structure of the food product [5,6]. Therefore, it is crucial to find a safe method to delay the browning and deterioration of *A. bisporus*.

Lactic acid bacteria (LAB) are generally recognized as safe (GRAS) by the Food Drug Administration (FDA) and have a positive effect on the preservation of agricultural products [7,8,9], which are applied to inhibit the growth of spoilage microorganisms, delay peel discoloration, regulate the metabolism of reactive oxygen species, and decompose the browning substrates. It was found that *Lactobacillus delbrueckii* screened from fermented yak milk effectively extended the shelf life of strawberries by inhibiting the growth of spoilage microorganisms [10]. In addition, a *Lactiplantibacillus plantarum* (*Lp. plantarum*) suspension controlled the postharvest peel discoloration in litchi [11]. *Lp. plantarum* also protected silage from lipid peroxidation by improving the antioxidant capacity [12]. Microbes have enzyme systems that break down and convert phenolic substances, in which *Lp. plantarum* can break down and convert protocatechuic acid [13]. Moreover, the phenolic substances identified in *A. bisporus* include gallic acid, protocatechuic acid, catechin, and caffeic acid [14]. These phenolics combine and interact with the PPO contained in the tissues of the fruiting bodies and eventually oxidize to melanin. Melanin was deposited on the surface of the mushroom, and browning occurred. Although LAB have been studied to control spoilage microorganisms and the color of fruit and vegetables, the screening of browning-resistant LAB based on the characteristics of *A. bisporus* and their application to postharvest preservation have not yet been reported.

In the previous experiments, we screened and isolated 72 strains of LAB from Tibetan plateau fermented milk [15]. In this paper, we selected suitable substrates for PPO based on the typical phenolic compounds of *A bisporus*, (1) determined the interaction between 72 strains of LAB and phenolic compounds, (2) evaluated the ability of LAB to convert phenolics, (3) screened the strains with the highest sensory scores for applications to the browning resistance and preservation of *A. bisporus*, (4) evaluated the spoilage microbial and reactive oxygen metabolism of *A. bisporus* during storage, and (5) systematically evaluated the effect of optimal strains on browning and quality deterioration of *A. bisporus*.

## 2. Materials and Methods

### 2.1. Materials

Seventy-two strains of LAB were isolated from traditional fermented dairy that was collected in Tibetan Autonomous Prefecture, Gansu Province, in May 2021. *A. bisporus*, strain AS2796, were harvested from the Linze County edible mushroom cultivation base (Gansu, China). A total of 35 ± 5 g of mushrooms with good fruiting bodies was selected and precooled in cold storage (4 °C and 90% Relative Humidity) for 24 h.

### 2.2. LAB Sensitivity to Phenolic Compounds

The Michaels–Menton constant (Km) reflects the affinity of the enzyme for the substrate, with reference to the value of Km for the enzymatic reaction of polyphenol oxidase with polyphenolic substances in *A. bisporus* [16]. Gallic acid, protocatechuic acid, and catechins present in *A. bisporus* were selected as the best substrates for PPO. Seventy-two strains of LAB were activated, and 1.5% of the bacteria solution was inoculated into the MRS liquid medium containing polyphenol standard. The concentration of phenolic compounds in the MRS medium was adjusted to 0.4 g L^−1^ or 0.8 g L^−1^. LAB was incubated at 37 °C, and an enzyme marker was used to measure the growth, following absorbance readings at 600 nm every 2 h.

### 2.3. Screening of Optimal Strains by Sensory Evaluation

The ability of LAB to convert phenolic compounds was characterized by high-performance liquid chromatography (HPLC) analysis. Gallic acid, protocatechuic acid, and catechin standards of 99% purity were diluted to 0.25 g L^−1^. Next, the LAB were mixed with a standard solution of phenolic compounds and incubated for 18 h, centrifuged (10,000× *g*, 5 min), and the supernatants were filtered and analyzed by HPLC.

HPLC (Agilent 1260 Infinity II, China Agilent Technologies Co. Ltd., Shanghai, China) was performed with a quadratic gradient pump, diode array detector, and Column symmetry C18 (250 mm × 4.6 mm, 5 μm). Mobile phase A was 0.05% acetic acid solution, mobile phase B was 0.05% acetonitrile, and autosampling was performed (0.05 µL). Optimized chromatographic conditions: temperature, 30 °C; linear gradients from 5% to 55% B for 10 min, 55% to 35% B for 12 min, 35% to 70% for 15 min, and 70% for 20 min applied at a flow rate of 0.8 mL min^−1^. The conversion rate was calculated using the peak area method and is expressed as a percentage (%):$$\mathrm{Conversion}\;\mathrm{rate}=\frac{Post\;conversion\;concentration}{Original\;standard\;concentration}$$

### 2.4. Screening of Optimal Strains

The LAB to be evaluated were sprayed on the mushrooms (1 mL of the bacterial solution per spray, 1.44 × 10^8^ CFU mL^−1^), and the samples were given sensory scores every 3 d during a 12 d storage period (4 ± 1 °C, RH 90%).

The sensory evaluation method has been referenced in a previously published paper, but with a modification [17]. Ten experts (five males and five females) conducted the scoring of the assessment after adequate rigorous training. The scoring criteria are listed in Table 1 (using a 0–20 score system). Fresh *A. bisporus* were taken as the standard (total score: 100).

### 2.5. Concentration of Lp. plantarum NML21 Used and Packaging Scheme for A. bisporus Samples

The enriched NML21 strain was dissolved in deionized water and quickly sprayed on the surface of mushrooms, with each spray containing approximately 1.44 × 10^8^ CFU of microorganisms [18]. The mushroom samples were divided into four groups, and each group had one hundred and thirty boxes (sterile polypropylene plastic boxes, 20 cm × 25 cm × 15 cm, each containing 8–10 mushrooms), with the control that went without treatment, while the remaining M1, M5, and M9 groups were sprayed once, five, and nine times, respectively. The mushrooms were air-dried for 45 min after treatment and stored for 18 d under the cold storage conditions. Samples were collected every 3 d to determine the relevant indicators.

### 2.6. Determination of Weight Loss, Firmness, Browning Index (BI), and Respiratory Rate

The mushrooms were weighed using an electronic scale, and the initial weight of the mushrooms was recorded as W_0_, which was the weight of each sampling day. The weight loss is expressed as a percentage (%), which was calculated using the following equation:$$\mathrm{Weight}\;\mathrm{loss}\;(\%)=\frac{\left(W_{0}-W_{1}\right)}{W_{0}}$$

Firmness was determined using texture profile analysis (TMS-2000, Weixun Super-technology Instrument Co., Beijing, China) by selecting P2 mode paired with a 2 mm diameter circular probe for the puncture testing of *A. bisporus.* From each experimental group, five mushrooms were randomly selected and tested equidistantly on the cap. Five hundred g weights were used to calibrate the weight. The unit of firmness is indicated as N. The test parameters were set as follows: puncture depth: 10 mm; test speed: 1 mm s^−1^; pretest speed: 1 mm s^−1^.

The BI was determined using the method of Palou et al. [19]. Three points were evenly selected for each mushroom cap for measurement. The BI (L, a, and b) was measured using a Ci6x (X-Rite, USA) portable spectrophotometer. BI was calculated as follows:(1)$$BI=\frac{100\left(x-0.31\right)}{0.17}$$
(2)$$x=\frac{\left(a+1.75L\right)}{\left(5.645L\right)+(a-3.012b)}$$

A fruit and vegetable respirometer (SY-1022) was used to determine the respiratory intensity of *A. bisporus*. A 500 g sample was placed in a 1 L respirometer container, a carbon dioxide sensor was inserted, and the respiration intensity of *A. bisporus* was measured for 3 min and averaged 5 times each time. The respiratory intensity is expressed as mg kg^−1^ h^−1^ CO_2_.

### 2.7. Counting Method of Pseudomonas *spp*.

The microorganisms were determined according to the method of Nasiri et al. [20], with a slight modification. Briefly, 15 g mushroom sample was pounded and mixed well with 135 mL of 0.1% peptone water and diluted in a gradient (from 10^−1^ to 10^−9^). A total of 0.2 mL dilution was taken and spread evenly throughout the selection medium, and the culture was counted as required. Cephaloidin fuccin cetrimide medium was used to culture *Pseudomonas* spp. under these growth conditions for 2 d at 25 °C. Viability was expressed as the log of colony-forming units per gram (log CFU g^−1^).

### 2.8. Activities of PPO, Superoxide Dismutase (SOD), Catalase (CAT), and Levels of Hydrogen Peroxide (H_2_O_2_) and Superoxide Anion (O^2.-^) in Mushrooms

The enzyme extract was prepared before evaluating its efficacy as previously described [21]. A total of 3 g of chopped mushroom tissue was ground in an ice bath, and 6 mL of 0.05 M phosphoric acid buffer (contains 5% (*m*/*v*) PVP and 0.01 (*v*/*v*) Triton X-100) at pH 6.8 was added during the regrinding process. The ground samples were centrifuged at 10,000× *g* for 20 min at 4 °C. The supernatant was the crude enzyme solution.

The PPO enzyme activity was determined with reference to the method of Augustin et al. with slight modifications [21]. The determination system of PPO consisting of 4.8 mL phosphoric acid buffer at pH 6.8 and 1 mL 0.05 M catechol solution was mixed in a 20 °C, constant-temperature water bath. A total of 0.2 mL phosphoric acid buffer was added to the blank, while the treated sample group contained 0.2 mL enzyme extract. The OD value was immediately measured at 420 nm, and the change in OD value was recorded every 20 s for 2 min. SOD activity was measured following the approach of Liu et al., and the CAT activity was determined according to the method of Dong et al. [22,23]. The activities of the above are expressed as U kg^−1^ prot.

H_2_O_2_ accumulation was determined following the approach of Prochazkova et al. [24]. A standard curve was made with the standard solution of H_2_O_2_. Absorbance at 412 nm was detected, and the contents are displayed as mol kg^−1^.

The determination of O^2.-^ indicators was made according to the method of Fang et al. [25]; 3 g sample of mushroom was accurately weighed, ground in an ice bath with 5 mL of 0.05 mol L^−1^ phosphate buffer (pH 7.8), and centrifuged at 12,000× *g* for 20 min. Next, 1 mL of supernatant was taken, and 1 mL of 0.05 mol L^−1^ phosphoric acid buffer and 1 mL of 0.001 mol L^−1^ hydroxylamine hydrochloride solution were added, mixed, and incubated at 25 °C for 1 h. Then, 1 mL of 0.017 mol L^−1^ *p*-aminobenzenesulfonic acid and 1 mL of 0.007 mol L^−1^ α-naphthylamine were added to the sample and incubated for 20 min at 25 °C for a color development reaction. The absorbance value of the chromogenic solution was measured at 530 nm, and the measured value without incubation was used as a reference control. The unit of O^2.-^ production rate is indicated as mmol s^−1^ kg^−1^.

### 2.9. Determination of Cell Conductivity and Malondialdehyde Content (MDA)

The cell membrane conductivity was determined with reference to Xu et al. [26]. The experiment was performed by cutting 2 mm wide, round slices of mushroom tissues with a razor blade and immersing them in 25 mL of deionized water. Conductivity (P1) was measured with a conductivity meter (DDSJ-308A, Shanghai Yidian Analytical Instruments Co., Shanghai, China) after 30 min of holding the samples in a water bath at 30 °C. The solution was boiled for 5 min, cooled to about 25 °C, and then conductivity was measured again as P2. The equation is as follows:$$Relative\;conductivity\left(\%\right)=\frac{P1}{P2}$$

The content of MDA was determined using the thiobarbituric acid method [27]. A total of 0.2 g of mushroom sample was ground in liquid nitrogen, and then 1.5 mL of 100 g L^−1^ trichloroacetic acid solution was added and mixed thoroughly. The supernatant was obtained after centrifugation at 10,000× *g* for 20 min at 4 °C. A total of 1 mL of 0.67% (*w*/*v*) thiobarbituric acid solution was added to 1 mL of the supernatant, mixed, and boiled for 20 min. The absorbances A_532_, A_600_, and A_450_ of the mixed system were measured at the 450 nm, 532 nm, and 600 nm wavelengths, respectively, and MDA content is expressed in nmol kg^−1^. The MDA content of sample was calculated according to the following equation:$$MDA={[(A}_{532}{-A}_{600}){\times 6.452-0.56\times A}_{450}]$$

### 2.10. Statistical Analysis

The sensory evaluation experiments lasted for 12 d. The optimal strain, *Lp. plantarum* NML21, was screened, and the other storage experiments lasted for 18 d. During the storage period, the samples were evaluated every three days, with three replications for each treatment. Correlation analysis used Pearson’s correlation for parametric analysis. The average and standard error (± SE) of the data were calculated using Microsoft Excel 2016 (*n* > 3). The significance analysis of Duncan’s multiple differences was performed by using SPSS 26. Values of *p* < 0.05 were considered statistically significant. Figures were created using Origin 2021.

## 3. Results

### 3.1. Interaction of Phenolic Compounds with LAB

The growth statuses of 72 LAB strains were studied in the culture media containing gallic acid, protocatechuic acid, and catechins, respectively. The representative data of 20 strains of LAB and phenolic compounds interactions were selected (Table 2). Polyphenols promoted the growth of strain Nos. 1–10; the OD value of strain No. 2 NML21 in 0.8 g L^−1^ gallic acid medium was 0.18 higher than that of the control. In contrast, polyphenols inhibited the growth of strain Nos. 11–20; the OD value of strain No. 19 YLJ2 in 0.8 g L^−1^ catechin medium was 0.77 lower than that of the control. The growth of 35 strains was not inhibited by the polyphenols substances. These strains were considered to have the potential to convert the browning substrate.

### 3.2. Determination of Phenolic Conversion Ability of LAB by HPLC

HPLC (peak area method) was further used to determine the conversion capacity. Six strains of LAB had a remarkable ability to convert the polyphenols. Among them, the conversion rates of gallic acid and protocatechuic acid by *Lp. plantarum* NML21 were as high as 97.16% and 95.85%, respectively (Figure 1). In addition, the remaining five strains had a conversion rate more than 45% for the three polyphenolic substances within the specified time (18 h). In conclusion, six strains gradually converted the phenolic compounds in the medium during growth.

### 3.3. Screening for the Optimal Strain to Use in the Treatment

Six strains of LAB (MN3, NML21, HG26, TG4, NX3, and HG23) were sprayed on *A. bisporus*. As shown in Table 3, with the extension of the storage time, the deterioration of *A. bisporus* occurred and the sensory scores gradually decreased. The overall quality condition of the mushrooms decreased markedly from 3 to 9 d. The sensory scores were significantly higher in the NML21 treatment compared with those of the other groups on 3, 6, 9, and 12 d (*p* < 0.05), with scores greater than 60 on 12 d. Strain NML21 had the best ability to delay postharvest quality deterioration of *A. bisporus.*

### 3.4. Effects of NML21 Treatments on the Overall Acceptability, BI, and Weight Loss of A. bisporus during Storage

The visual images, BI, and weight loss of *A. bisporus* play a crucial role in consumer purchase decisions. With a prolonged storage time, the acceptability of *A. bisporus* continuously decreased, and the browning and weight loss rates increased. The quality of *A. bisporus* could be judged visually (Figure 2). As shown in Figure 3A, the control deteriorated severely on 15 d, and the BI was as high as 82.7, while the treatment group M5 had only slight browning with the lowest BI (58.6). The BI of the remaining two treatment groups (M1 and M9) was also significantly lower than that of the control (*p* < 0.05). The weight loss rate was reduced in the treated group (Figure 3B). The treatment group M5 was significantly more affected than the other groups on 6 d (*p* < 0.05). The above results showed that treatment of M5 had a remarkable effect on the storage of *A. bisporus*, while reducing the BI and weight loss rate.

### 3.5. Effects of NML21 Treatment on Firmness and Respiration Rate of A. bisporus

Firmness directly reflects the degree of softness of *A. bisporus*. According to Figure 3C, the firmness of *A. bisporus* continued to decrease during the storage time. In comparison with the control, the decreasing trend of treatment group M5 was obviously smaller than those of M1 and M9, and the firmness of M5 (2.11 N) was significantly higher than the other sample groups on the 15th d of storage (*p* < 0.05). It is notable that the firmness of the treatment group M9 was lower than that of the control during storage. Treatment group M5 showed the best retention of firmness.

The respiration rate affects the metabolic rate of *A. bisporus* fruiting bodies. According to Figure 3D, *A. bisporus* has two respiratory peaks at 6 d and 15 d. The first respiration rate peak of M5 was 358.74 mg kg^−1^ h^−1^ CO_2_, and the respiration rate of M5 was significantly lower than those of the other sample groups on 6 d (*p* < 0.05). Treatment group M5 had a lower respiration rate.

### 3.6. Effects of NML21 Treatment on the Number of Pseudomonas *spp*. in A. bisporus

*Pseudomonas* spp. are the main microorganisms responsible for the spoilage and deterioration of *A. bisporus* during storage. The differences in microbial counts between the four treatment groups were slightly changed during the 0–3 d period (Figure 4). However, the differences between the groups increased after 6 d, with the number of *Pseudomonas* spp. in treatment group M9 being significantly lower than those of the remaining three groups (*p* < 0.05). The number of LAB sprays showed a negative correlation with the count of *Pseudomonas* spp. The above results illustrate that *Lp. plantarum* NML21 inhibited the growth of *Pseudomonas* spp.

### 3.7. Effects of NML21 Treatments on the Cellular Integrity of A. bisporus

MDA and cell conductivity could indicate the integrity of the tissue cells of *A. bisporus* and the aging and breakage of the cell membrane. Based on Figure 5, we found that the MDA content and cell conductivity both showed increasing trends during storage. The MDA content maintained a dynamic equilibrium state at 3–6 d, and then increased rapidly. The MDA values for the treatment M5 were significantly lower than those of the rest of the groups at 9–15 d (*p* < 0.05). The cell conductivity value of treatment group M9 was higher than those of the rest groups on 0–9 d, but those of the three treatment groups were smaller than that of the control on 9–15 d. The above results suggest that the treatment given to M5 was the best solution to reduce the degree of membrane lipid peroxidation and electrolyte leakage from the fruiting bodies during storage.

### 3.8. Effects of NML21 Treatment on the O^2.-^ Production Rate and H_2_O_2_ Content of A. bisporus

The O^2.-^ production rate and H_2_O_2_ content of *A. bisporus* reflect the intensity of tissue oxidative stress during storage. As shown in Figure 6A, the O^2.-^ production rates of all the treatments presented an increasing trend, and the O^2.-^ production rates of the treatment group M5 were all significantly lower than that of the control (*p* < 0.05). From Figure 6B, the H_2_O_2_ content increased rapidly in 3 d and exhibited a single peak shape at 3–6 d. Between 6 and 15 d, the H_2_O_2_ content of the treatment group M5 was significantly lower than those of the remaining three groups (CK, M1, and M9) (*p* < 0.05), reaching its lowest point at 9 d (1.97 × 10^−3^ mol kg^−1^). The treatment given to M5 effectively delayed aging in the tissue of *A. bisporus* and maintained the dynamic balance of the reactive oxygen species (ROS).

### 3.9. Effects of NML21 Treatment on PPO, SOD, and CAT in A. bisporus

PPO is a key enzyme for the browning reaction of *A. bisporus*. According to Figure 7A, we found that after 6 d, the treatment was able to more significantly inhibit the activity of PPO compared to that of the control (*p* < 0.05). On the 6th d, the PPO activity in the treatment group M5 was significantly reduced by 86.76 × 10^3^ U kg^−1^, 90.8 × 10^3^ U kg^−1^, and 70.4 × 10^3^ U kg^−1^ compared to those of the CK, M1, and M9 groups, respectively (*p* < 0.05).

CAT and SOD can remove H_2_O_2_ and O^2.−^. From Figure 7B, the CAT activity increased slightly on 0–3 d and decreased gradually from 3–15 d. The CAT activity level of treatment group M5 was significantly higher than those of the other groups (CK, M1, and M9) at 5.58×10^3^ U kg^−1^ on 15 d (*p* < 0.05). Figure 7C presents the data on SOD activity. The SOD activity level increased in the M5 and M9 treatment groups and remained high for 6–9 d, which was significantly higher than that of the control (*p* < 0.05). In summary, the NML21 treatment could improve the scavenging ability of ROS, increase the CAT and SOD activity levels, and inhibit the activity of PPO.

### 3.10. Correlation Analysis of the Effects of NML21 Treatment on Storage Quality of A. bisporus

Pearson correlation analysis was used to measure the strength of the linear relationship between the two variables. The BI of *A. bisporus* showed a highly positive correlation with weight loss, the *Pseudomonas* spp. count, MDA content, cell conductivity, H_2_O_2_ content, O^2.−^ production rate, and PPO activity (Figure 8). The PPO activity showed a significantly negative correlation with the firmness of the fruiting bodies (*p* ≤ 0.01). Browning was weakly correlated with the respiration rate. These results show the interaction between the treatment given to M5 on the quality, ROS, and different enzyme activities during the storage of *A. bisporus*.

## 4. Discussion

Our study explored the possibility of *Lp. plantarum* as a new browning-resistant preservation technology for edible mushrooms. During postharvest storage and transportation, the cellular integrity of *A. bisporus* is disrupted by oxidative stress or external factors. The cellular contents (polyphenols and PPO) react with oxygen and further undergo a browning reaction, resulting in pigment accumulation [27]. At present, controlling browning in white mushrooms is mainly based on reducing tissue cell breakage or inhibiting PPO activity, but this approach only delays the appearance of the pigment and does not effectively reduce the final browning extent [28]. Therefore, the conversion of browning substrates is a new attempt. In our current research, the polyphenol substrate promoted or inhibited the growth of LAB (Table 2). This is similar to the findings previously reported, in which *Lactobacillus rhamnosus*, *Lp. plantarum*, and *Lactobacillus acidophilus* were promoted or inhibited growth in plant polyphenols [29]. Filannino et al. also found that gallic acid has a positive effect on the growth of LAB [30]. These findings indicate that the screened LAB utilized the phenolic compounds in the medium in their growth metabolism. To further confirm whether LAB converted the phenolic compounds, we used HPLC to determine the conversion. As expected, NML21 converted gallic acid, catechins, and protocatechuic acid into other substances (Figure 1). The previous research on LAB has shown a system of enzymes that breaks down phenolic compounds; its conversion capacity is mainly based on the 3-oxoadipats pathway [31]. Therefore, we speculate that catechin could be converted to epigallocatechin-3-gallate by *Lp. plantarum*, and then it could directly bind to the catalytic domain of PPO and compete with the substrate [32]. Nonetheless, the conversion mechanism of phenolic compounds by LAB remains to be further revealed.

Postripening occurs during the postharvest storage of *A. bisporus*, which is associated with the weight loss rate, firmness, and respiration rate [33]. In this study, the treatment given to M5 reduced the weight loss rate, maintained firmness, and delayed the appearance of a respiratory peak in *A. bisporus*. The results are similar to the report that LAB (*Streptococcus thermophilus* STCC0038 and *Lp. plantarum* STCC0007 mixture) reduced the weight loss rate, maintained firmness, and delayed the respiratory peak in lotus roots [34]. However, it is interesting to know that there was a clear dose–effect relationship in this result, with a higher BI, weight loss rate, and respiration rate, and more softening in the treatment group M9 than those in the control during the 0–6 d period (Figure 3). We infer that organic acids and hydrogen peroxide produced by LAB are toxic to mushroom fruiting bodies at high concentrations [35]. *Pseudomonas* spp. is the dominant spoilage microorganism in mushrooms. Tolaasin and the white line-inducing principle (extracellular substance) produced by *Pseudomonas* spp. are the main factors that induce mushroom discoloration and membrane ruptures, which are also involved in the activation process of PPO [36]. In the present study, the growth of *Pseudomonas* spp. was inhibited in treatment groups M5 and M9, which were partially consistent with the results that LAB (*Lactobacillus gasseri* LGchen and *Pediococcus pentosaceus* S-5-6) can antagonize *Pseudomonas* spp. [37]. *Pseudomonas* spp. could resist external stimuli through a biofilm, while the byproducts produced by LAB can inhibit the formation of a *Pseudomonas* spp. biofilm [38]. This is a correlation in biofilm inhibition with the AI-2/LuxS system of LAB. LAB suppressed the expression of the PAO1 signaling molecule AHL and inhibited elastase activity and reduced biofilm formation in *Pseudomonas* spp. [39]. In addition, the decrease in *Pseudomonas* spp. may also be due to the influence of NML21, causing an alteration in the nutritional niche or nutritional competition [40].

The disturbance of the reactive oxygen metabolism system is an important cause of the aging and deterioration of *A. bisporus* during storage. Under steady-state conditions, the reactive oxygen metabolism of *A. bisporus* is in dynamic equilibrium [41]. However, the antioxidant capacity of *A. bisporus* gradually weakened over time, unable to effectively scavenge excess ROS, leading to oxidative stress and cell structure and damage due to free radicals [42]. In this study, the treatment given to M5 reduced the number of ROS, controlled cell membrane damage, and increased the CAT and SOD activity levels in *A. bisporus*. This is similar to the findings previously reported in which a microorganism (*Cryptococcus laurentii*) increased the activities of SOD and CAT in tomato fruit [43]. Alejandra et al. found that *Lactococcus lactis* A-NZ9000 (pVE3655) produced CAT, which could be the reason for the higher CAT activity level than that of the control [44]. We also speculated that exogenous NML21, as an external stressor, continuously stimulated the ROS scavenging system of *A. bisporus*. To further confirm whether ROS were scavenged in *A. bisporus*, H_2_O_2_ and O^2.−^ were measured in this study, and we found that the ROS were effectively scavenged, and this result affirms that an appropriate concentration of NML21 treatment had a positive effect on the homeostatic maintenance of ROS in *A. bisporus*. Cellular lipid membrane oxidation and electrolyte leakage are specific manifestations of ROS attacking the cells. The results of the MDA content and cell conductivity also confirmed that the NML21 treatment reduced the accumulation of free radicals in the cells [45]. However, whether NML21 is directly involved in the regulation of CAT and SOD activities needs to be further investigated.

Correlation analysis showed that the browning of *A. bisporus* was significantly and positively correlated with the MDA content and the number of *Pseudomonas* spp. (*p* < 0.05), which suggests that tissue cell membrane damage is the most important causative factor for browning. The study found that the metabolite of LAB, γ-aminobutyric acid (GABA), delayed the browning of *A. bisporus*, promoted PAL gene expression, and inhibited PPO gene expression [46]. Therefore, we hypothesized that the secondary metabolites of NML21 acted as signaling molecules to directly activate the ROS scavenging system and attenuate the ROS-induced cell membrane damage. PPO activity was significantly negatively correlated with firmness (*p* < 0.01), probably due to the softening of the fruiting bodies and the release of PPO from inside the cells. In addition, the weak correlation between browning and the respiration rate might indicate that low-temperature storage (4 °C) suppressed the metabolic rate of the fruiting bodies. Although NML21 inhibited the browning of *A. bisporus*, the metabolites of the strain are complex. The interaction between the mushroom, strain, and metabolites needs to be further clarified. The possible pattern of *Lp. plantarum* NML21 inhibiting browning, mitigating oxidative damage, and scavenging reactive oxygen species in *A. bisporus* is illustrated in the Graphical Abstract (using Figdraw ID:IWWIT4aafa).

## 5. Conclusions

The optimal substrates for PPO in *A. bisporus* were identified as gallic acid, catechin, and protocatechuic acid, and a browning-resistant strain, *Lp. plantarum* NML21, was selected from 72 strains for the preservation of *A. bisporus*.

After 15 d of storage, the treatment given to M5 reduced the BI, weight loss rate, respiration rate, firmness, and inhibited the growth of *Pseudomonas* spp. in the *A. bisporus* samples. In addition, the treatment given to M5 increased the SOD and CAT activity levels, inhibited PPO activity, controlled the accumulation of reactive oxygen species (O^2.−^ and H_2_O_2_), and reduced MDA content and cell conductivity. The above results confirmed that NML21 alleviated the discoloration and postripening effects of *A. bisporus*, controlled microbial-induced spoilage, and attenuated electrolyte leakage from the substrates by activating the reactive oxygen metabolism system of *A. bisporus*. Compared with the control, the *A. bisporus* in treatment group M5 presented a higher commercial value. In view of the effect of NML21 on the postharvest preservation of *A. bisporus*, it can be considered for development as a biopreservative.

## Figures and Tables

**Figure 1 foods-13-00168-f001:**
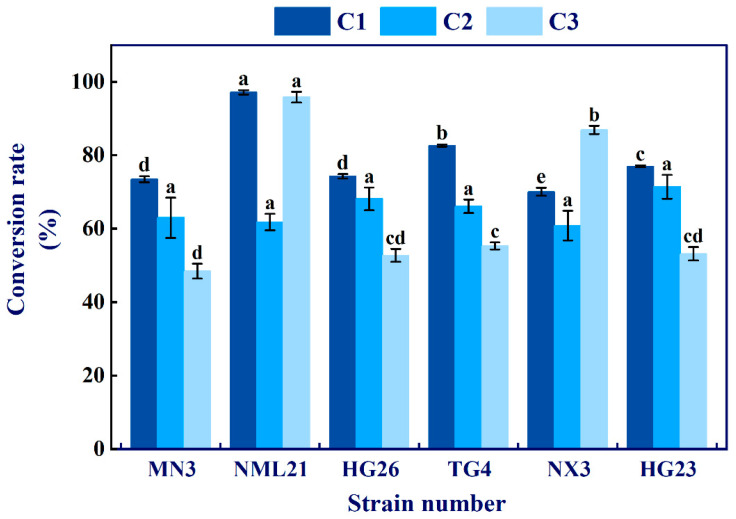
The conversion of polyphenols by LAB. C1, C2, and C3 represent the conversion rates of gallic acid, catechin, and protocatechuic acid, respectively. The bars represent the standard error (±SE). The different letters represent significant differences between the groups (*p* < 0.05).

**Figure 2 foods-13-00168-f002:**
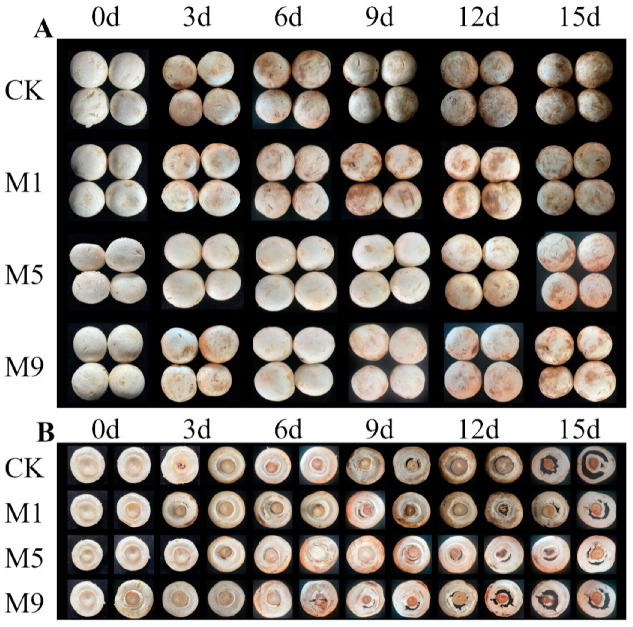
Effects of *Lp. plantarum* NML21 treatments on sensory quality of *A. bisporus*. (**A**,**B**) represent the front and back of the mushroom, respectively. CK represents the control (no treatment), and M1, M5, and M9 represent those sprayed once, five, and nine times, respectively, with NML21 standard solution. The same is described below.

**Figure 3 foods-13-00168-f003:**
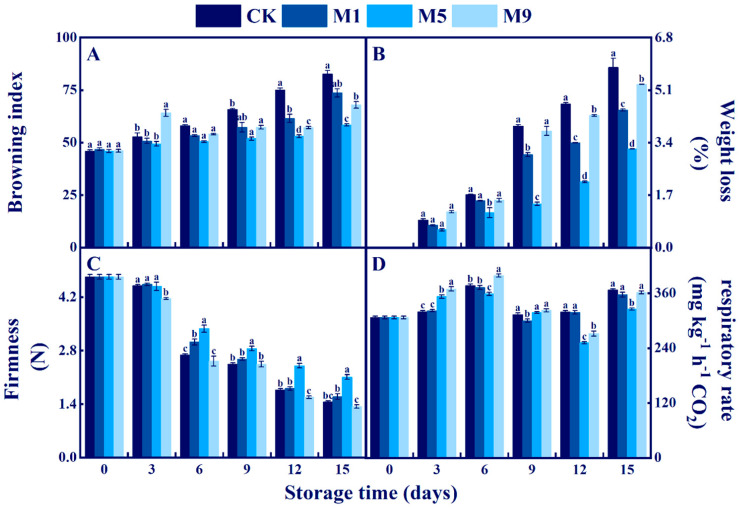
Effects of *Lp. plantarum* NML21 treatments on BI (**A**), weight loss (**B**), firmness (**C**), and respiratory rate (**D**) of *A. bisporus*. CK represents the control (no treatment), and M1, M5, and M9 represent those sprayed once, five, and nine times, respectively, with NML21 standard solution. Bars indicate standard error (±SE). Different letters represent significant differences within groups (*p* < 0.05).

**Figure 4 foods-13-00168-f004:**
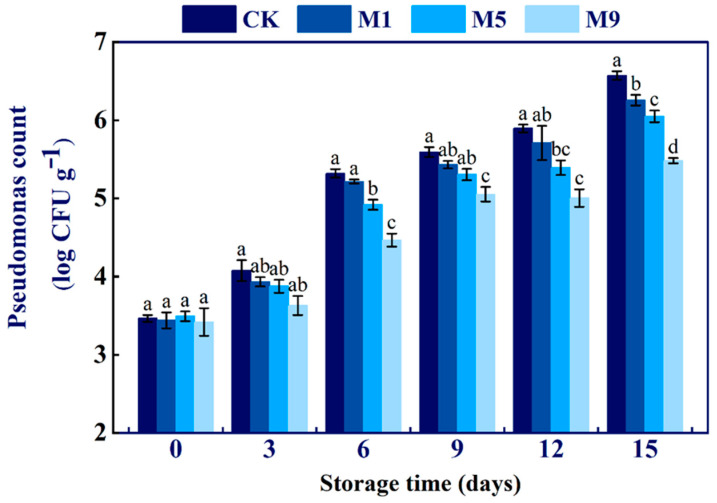
Effects of *Lp. plantarum* NML21 treatments on the count of *Pseudomonas* spp. CK represents the control (no treatment), and M1, M5, and M9 represent those sprayed once, five, and nine times, respectively, with NML21 standard solution. Bars indicate standard error (±SE). Different letters represent significant differences within groups (*p* < 0.05).

**Figure 5 foods-13-00168-f005:**
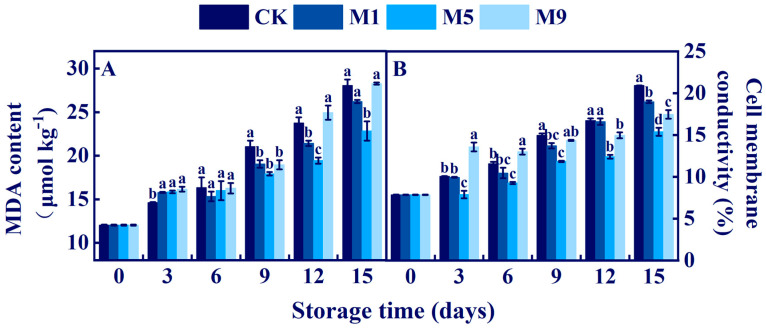
Effects of *Lp. plantarum* NML21 treatments on the MDA (**A**) content and cell conductivity (**B**). CK represents the control (no treatment), and M1, M5, and M9 represent those sprayed once, five, and nine times, respectively, with NML21 standard solution. Bars represent standard error (±SE). Different letters represent significant differences within groups (*p* < 0.05).

**Figure 6 foods-13-00168-f006:**
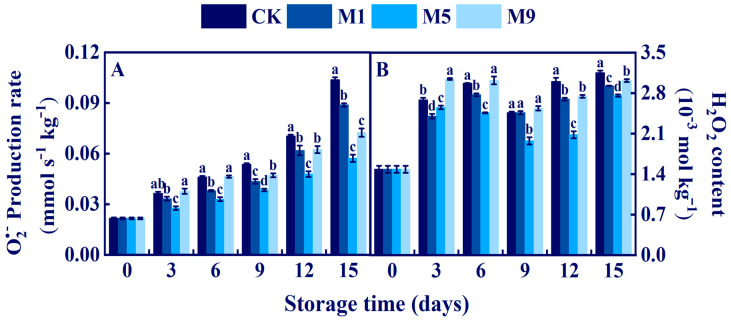
Effects of *Lp. plantarum* NML21 treatments on the O2^.−^ production rate (**A**) and H_2_O_2_ content (**B**) of *A. bisporus.* CK represents the control (no treatment), and M1, M5, and M9 represent those sprayed once, five, and nine times, respectively, with NML21 standard solution. Bars represent standard error (±SE). Different letters represent significant differences within groups (*p* < 0.05).

**Figure 7 foods-13-00168-f007:**
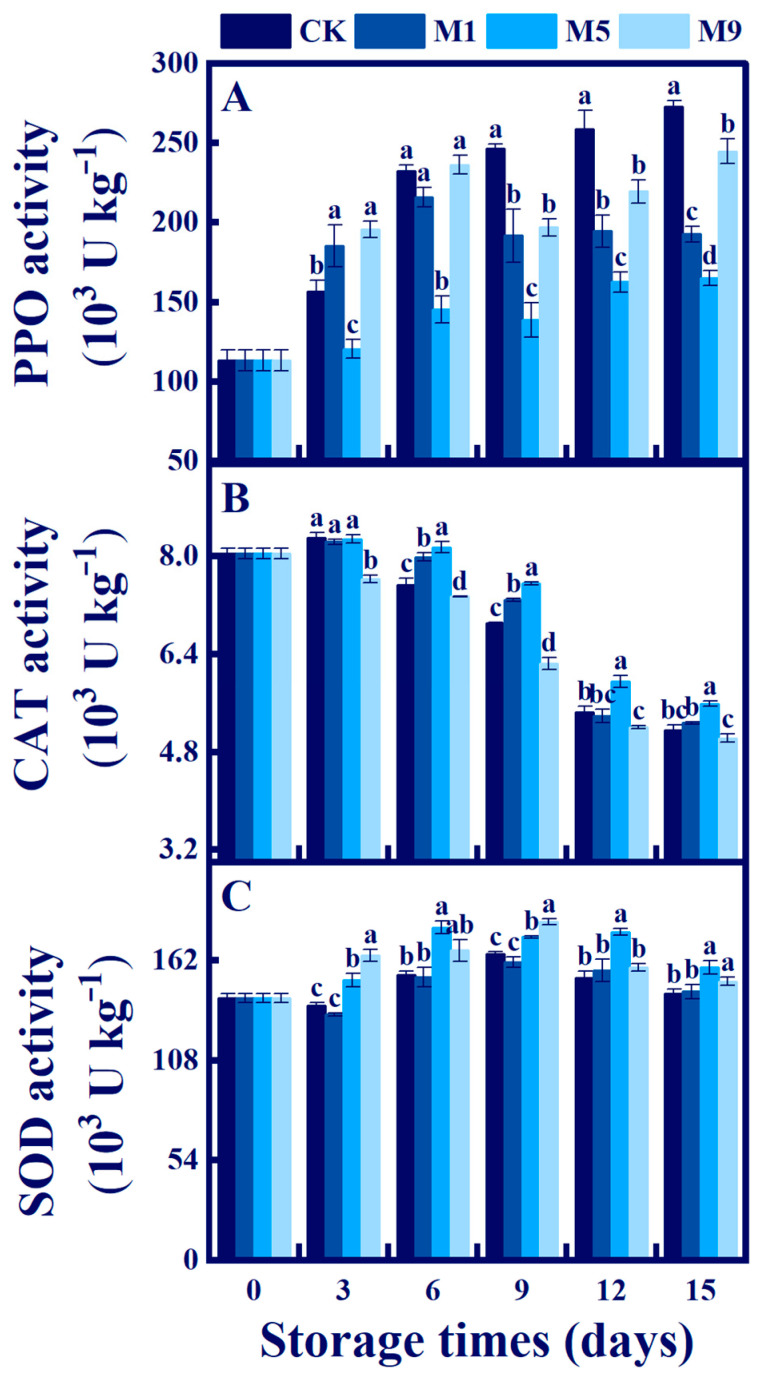
Effects of *Lp. plantarum* NML21 treatments on PPO (**A**), SOD (**B**), and CAT (**C**) in *A. bisporus*. CK represents the control (no treatment), and M1, M5, and M9 represent those sprayed once, five, and nine times, respectively, with NML21 standard solution. Bars indicate standard error (±SE). Different letters represent significant differences within groups (*p* < 0.05).

**Figure 8 foods-13-00168-f008:**
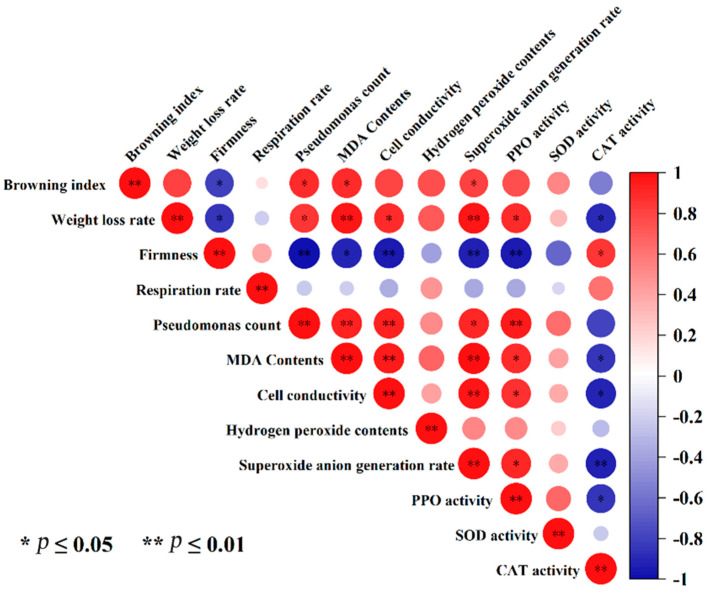
Correlation analysis of the effects of *Lp. plantarum* NML21 treatments on storage quality of *A. bisporus*.

**Table 1 foods-13-00168-t001:** *A. bisporus* sensory evaluation standard.

Score	Browning Degree (S1)	Cap Morphology (S2)	Smell (S3)	Softness of Fruiting Body (S4)	Consumer Acceptance (S5)	Sensory Score
20–16	White and lustrous	Closed/No dent	Clear aroma of *A. bisporus*/No LAB flavor	Stretchy	Very Satisfied	
15–11	Slight browning	Slightly open/Few dents	*A. bisporus* fragrance/Slight LAB odor	Slightly soft	Satisfied	
10–7	Mild browning	Half open/More dents	No distinctive fragrance of *A. bisporus*/LAB odor is obvious	Mildly soft	Grudging acceptance	
<7	Seriously browning	Totally open/Severe denting	Severely Off-odor/LAB odor is strong	Severely soft	Unacceptable	

Here are five scoring sections listed in the table, each scored out of 20, and the five sections total 100 points. With a score of less than 60, we considered the mushroom to be unacceptable to the consumer and without commercial value.

**Table 2 foods-13-00168-t002:** Growth response of 20 LAB strains at the presence of different phenols.

Strains Number	Control	Gallic Acid		Catechin		Protocatechuic Acid	
		0.4 g L^−1^	0.8 g L^−1^	0.4 g L^−1^	0.8 g L^−1^	0.4 g L^−1^	0.8 g L^−1^
1-MN3	1.48 ± 0.13	1.42 ± 0.05	1.59 ± 0.11	1.68 ± 0.07	1.72 ± 0.05	1.49 ± 0.04	1.49 ± 0.10
2-NML21	1.53 ± 0.09	1.65 ± 0.10	1.71 ± 0.14	1.55 ± 0.09	1.58 ± 0.13	1.64 ± 0.10	1.59 ± 0.06
3-HG26	0.87 ± 0.01	1.18 ± 0.04	0.77 ± 0.13	1.28 ± 0.10	1.23 ± 0.02	1.22 ± 0.07	1.30 ± 0.06
4-TG4	0.67 ± 0.14	1.05 ± 0.05	1.41 ± 0.03	1.23 ± 0.04	1.53 ± 0.02	1.46 ± 0.09	1.41 ± 0.12
5-NX3	1.54 ± 0.08	1.66 ± 0.03	1.43 ± 0.13	1.60 ± 0.15	1.64 ± 0.07	1.52 ± 0.03	1.65 ± 0.08
6-HG23	1.60 ± 0.12	1.87 ± 0.01	1.81 ± 0.08	1.58 ± 0.07	1.35 ± 0.01	1.55 ± 0.12	1.63 ± 0.15
7-XWW1	1.67 ± 0.03	1.34 ± 0.03	1.02 ± 0.05	1.32 ± 0.12	0.85 ± 0.03	1.42 ± 0.11	1.39 ± 0.09
8-Q10	1.66 ± 0.05	1.65 ± 0.01	1.62 ± 0.13	1.51 ± 0.07	1.63 ± 0.10	1.52 ± 0.02	1.62 ± 0.03
9-ZHG1	0.92 ± 0.04	1.64 ± 0.02	1.32 ± 0.07	1.25 ± 0.07	1.33 ± 0.06	0.92 ± 0.07	1.34 ± 0.07
10-NML22	1.69 ± 0.08	1.67 ± 0.15	1.69 ± 0.11	1.55 ± 0.09	1.54 ± 0.11	1.76 ± 0.03	1.72 ± 0.01
11-HG1	1.43 ± 0.01	1.01 ± 0.05	0.52 ± 0.03	1.23 ± 0.14	1.31 ± 0.01	1.42 ± 0.09	1.37 ± 0.08
12-XH1	1.51 ± 0.02	1.40 ± 0.13	1.39 ± 0.07	1.50 ± 0.08	1.45 ± 0.11	1.32 ± 0.05	1.27 ± 0.09
13-GN3	1.55 ± 0.03	1.53 ± 0.02	1.45 ± 0.13	1.32 ± 0.07	1.30 ± 0.08	1.48 ± 0.11	1.02 ± 0.11
14-TG1	1.63 ± 0.03	1.54 ± 0.06	1.37 ± 0.09	1.62 ± 0.03	1.53 ± 0.04	1.51 ± 0.01	1.42 ± 0.04
15-TG2	1.59 ± 0.04	1.41 ± 0.13	1.63 ± 0.08	1.42 ± 0.04	1.39 ± 0.09	1.41 ± 0.14	1.33 ± 0.10
16-XH1	1.62 ± 0.06	1.60 ± 0.03	1.70 ± 0.02	1.26 ± 0.04	0.68 ± 0.11	1.02 ± 0.12	1.01 ± 0.05
17-XWW2	1.54 ± 0.09	1.60 ± 0.01	1.47 ± 0.02	1.33 ± 0.09	0.92 ± 0.07	1.52 ± 0.03	1.43 ± 0.12
18-YLJ1	1.52 ± 0.09	1.48 ± 0.14	1.31 ± 0.02	1.48 ± 0.10	1.20 ± 0.01	1.34 ± 0.14	1.22 ± 0.08
19-YLJ2	1.10 ± 0.12	0.84 ± 0.04	0.79 ± 0.09	0.72 ± 0.10	0.33 ± 0.13	1.32 ± 0.06	1.42 ± 0.08
20-CQ5	1.54 ± 0.10	1.27 ± 0.09	1.38 ± 0.13	1.36 ± 0.06	1.38 ± 0.08	1.44 ± 0.11	1.40 ± 0.03

Twenty strains of LAB were grown in high or low (0.4 g L^−1^ and 0.8 g L^−1^) concentrations of gallic acid, catechin, and protocatechuic acid, respectively. Independent experiments were performed for each strain. Control group: the basal medium was only used for the same strain. Experimental group: the concentration and type of polyphenols were controlled under the premise of using basic medium. The data represent the absorbance values (OD: 600 nm).

**Table 3 foods-13-00168-t003:** Effects of LAB treatments on the sensory scores of *A. bisporus*.

		Time		0 d	3 d	6 d	9 d	12 d
Index								
Sensory score	CK			100 ± 0.00	92.00 ± 0.23 ^b^	80.18 ± 0.72 ^c^	60.63 ± 0.47 ^cd^	51.43 ± 0.67 ^d^
	Strain name		MN3	100 ± 0.00	91.11 ± 0.94 ^b^	82.32 ± 1.03 ^c^	59.97 ± 0.63 ^d^	59.08 ± 0.44 ^c^
			NML21	100 ± 0.00	91.64 ± 0.44 ^b^	90.56 ± 0.47 ^a^	73.43 ± 1.31 ^a^	65.58 ± 0.49 ^a^
			HG26	100 ± 0.00	92.68 ± 0.84 ^b^	81.91 ± 0.53 ^c^	64.42 ± 0.71 ^b^	61.17 ± 0.14 ^c^
			TG4	100 ± 0.00	95.73 ± 0.62 ^a^	86.20 ± 0.51 ^b^	62.20 ± 0.57 ^bcd^	64.15 ± 0.98 ^ab^
			NX3	100 ± 0.00	89.02 ± 0.85 ^c^	76.89 ± 0.94 ^d^	62.60 ± 0.49 ^bc^	52.03 ± 1.74 ^d^
			HG23	100 ± 0.00	95.22 ± 0.80 ^a^	91.62 ± 0.96 ^a^	72.25 ± 0.48 ^a^	61.66 ± 0.44 ^bc^

The optimal strain was screened by sensory evaluation. With a score of less than 60, we considered the mushrooms to be unacceptable to the consumer and without commercial value. MN3, NML21, HG26, TG4, NX3, and HG23 are strains. The control group (CK) was not treated. The error margins represent the standard error (±SE). The different letters represent significant differences between the groups (*p* < 0.05).

## Data Availability

Data is contained within the article.

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
