# Peer review of "Screening of Lactiplantibacillus plantarum NML21 and Its Maintenance on Postharvest Quality of Agaricus bisporus through Anti-Browning and Mitigation of Oxidative Damage"

_foods, 2024, doi:10.3390/foods13010168_

Round 1
Reviewer 1 Report
Comments and Suggestions for Authors
This paper is interesting by screening and showing the biopreservation effect of some L. plantarum strains on the postharvest quality of Agaricus bisporus through a series of experimental data, and the potential use of a selected strain (LP NML21) for such an application. Despite of these, the manuscript contains a lot of imperfection to be corrected and improved, as listed mainly in the following table:
|
Line/§/section/ |
Errors |
Corrections/suggestions (in italic) |
|
Title |
Lactiplantibacillus plantarum NML21 |
Lactiplantibacillus plantarum NML21 |
|
Line 30 |
Resists browning |
Browning ? or anti-browning ? |
|
Line 35 |
Mushrooms[1] |
Mushrooms [1] |
|
Line 35-39 |
However, it lacks protective structures and highly susceptible to mechanical damage and pathogen infection after harvesting, which results in the autolysis of tissue cells and the destruction of the original cellular compartments. Thus initiating the enzymatic browning from the interaction of phenolic com- 38 pounds and polyphenol oxidase (PPO) in the cell contents[2]. |
Re-write for example: However, it lacks protective structures and is highly susceptible to mechanical damage and pathogen infection after harvesting. These results in the autolysis of tissue cells and the destruction of the original cellular compartments, which initiate the enzymatic browning from the interaction of phenolic com- 38 pounds and polyphenol oxidase (PPO) in the cell contents [2]. |
|
Line 48 |
FDA |
Food Drug Administration (FDA) |
|
Line56-58 |
Microbes have enzyme systems that break down and convert phenolic substances, in which Lp. plantarum can break down and convert pro tocatechuic acid[13 |
Microbes have enzyme systems that break down and convert phenolic substances. For example, Lp. plantarum can break down and convert protocatechuic acid [13
Suggestion: Explain the role of phenolic substances, PPO in browning. It is not clearly announced. |
|
Line 63-70 |
|
Please put space before and after all brackets |
|
Line 66 |
(2)evaluated of the ability of LAB to convert phenolics |
(2) evaluated the ability of LAB to convert phenolics |
|
Line 77 |
35±5 g |
35 ± 5 g |
|
Line 78 |
(4℃ and 90 RH) |
What is RH? Relative humidity? RH 90%? |
|
Line 79 |
LAB sensitivity to phenolic compounds was evaluated |
LAB sensitivity to phenolic compounds |
|
Line 80 |
Reference to the interaction of phenolic compounds of A. bisporus with PPO (Michaelis-Menten equation)[16]. |
The interaction of phenolic compounds of A. bisporus with PPO is determined according to (Michaelis-Menten equation)?? |
|
Line 86 |
600 after every 2h |
600 nm every 2h |
|
Line 96 |
0.05 ul |
0.05 µL ? |
|
Line 101 |
Screening of optimal strains |
Screening of optimal strains by sensory evaluation |
|
Line 102 |
Screening of optimal strains by sensory evaluation |
Delete this |
|
Line 110 table 1 |
|
Uniformize the spelling of “odor” or “odour” |
|
Line 114 |
Concentration of Lp. plantarum NML21 used and packaging scheme for A. bisporus samples |
Concentration of Lp. plantarum NML21 used and packaging scheme for A. bisporus samples |
|
Line 137 |
BI calculated the following |
BI was calculated as follow: |
|
Line 145 |
Counting method of Pseudomonas spp |
Counting method of Pseudomonas spp |
|
Line 192 |
inliquid |
in liquid |
|
Line 197 |
Sample was calculated… |
The MDA content of sample… |
|
Line 200 |
Statistical analysis |
Not clear. Rewrite |
|
Line 202 |
Lactiplantibacillus plantarum NML21 (Lp. plantarum NML21) |
Lp. plantarum NML21 |
|
Line 219 |
Footnote of the Table 2 * refers to what? Control corresponds to t=0? |
Please explain |
|
Line 224 |
had a remarkable ability to conversion of the polyphenols. |
had a remarkable ability to the conversion of the polyphenols. |
|
Line 242 |
Table 3 |
Interpretation: If score less than 60 is unacceptable, the control is still acceptable after 9 days? Application of LAB interesting if storage superior to 9 days? |
|
Line 258 |
As shown Figure. 3A |
As shown in the Figure 3A |
|
Line 278 |
The first respiration peak respiration rate |
The first respiration rate peak |
|
Line 279-280 |
Treatment M5 should more reduction in the respiration rate? |
Revise |
|
Line 281 |
Effects of NML21 treatment on the number of Pseudomonas spp. in A. bisporus |
Effects of NML21 treatment on the number of Pseudomonas spp. in A. bisporus |
|
Line 284 |
were little changed during the 0-3 d period |
were slightly changed during the 0-3 d period |
|
Line 303 |
groups(p |
groups (p |
|
Line 293 |
Figure. 5 |
Figure 5 Please correct in the manuscript (Delete point after Figure) |
|
Line 293-294 |
Based on Figure. 5, we found that the MDA content and cell conductivity both showed increasing trend. |
Based on Figure 5, we found that the MDA content and cell conductivity both showed increasing trend during the storage. |
|
Line 295 |
The treatment M5 was significantly lower than the rest of the groups from 9-15 d (p<0.05) |
The MDA values for the treatment M5 were significantly lower than the rest of the groups from 9-15 d (p<0.05) |
|
Line 304 |
Effects of NML21 treatment on the O2.- production rate and H2O2 content of A. bisporus |
Effects of NML21 treatment on the O2.- production rate and H2O2 content of A. bisporus |
|
Line 318 |
PPO is a rate-limiting enzyme for the browning reaction of A. bisporus |
Explain rate-limiting enzyme? |
|
Line 320 |
to the control(p<0.05) |
to the control (p<0.05) |
|
Line 323 |
Figure. 7B |
Figure 7B Please delete points after the word Figure in the manuscript. |
|
Line 331 |
A. bisporus |
A. bisporus |
|
Line 334-342 |
3.10 Correlation analysis section… |
Introduce briefly the correlation analysis method before announcing the results. |
|
Line 354 |
Lactobacillus rhamnosus, Lp. plantarum, and Lactobacillus acidophilus |
Lactobacillus rhamnosus, Lp. plantarum, and Lactobacillus acidophilus |
|
Line 360 |
(Figure. 2) |
Figure 1??? |
|
Line 360 |
Previous research on LAB has a system of enzymes that break down phenolic compounds |
Previous research on LAB has shown a system of enzymes that breaks down phenolic compounds |
|
Line 374 |
We infer that organic acids and hydrogen peroxide produced by LAB were toxic to mushroom fruiting body at high concentrations |
Why BI is higher, although hydrogen peroxide also inhibits PPO and limits browning? Is there any explanation? |
|
Line 379-380 |
In the present study, the growth of Pseudomonas was inhibited in treatment M5 and M9 |
Is there other mechanism of inhibition of Pseudomonas? Production of ROS? |
|
Line 398 |
Lactococcus slactis |
Lactococcus lactis |
1. There are some incomplete phrases (e.g., without verb).
2. Some sentences are not clear and need revision.
Author Response
Thank you so much for your comments. Please see the attachment.

Reviewer 2 Report
Comments and Suggestions for Authors
The manuscript describes an interesting study aimed at investigating whether a strain of L. plantarum can prevent the deterioration of Agaricus bisporus during storage. The methodology used is adequate, mostly, the English is good and the results are encouraging. It has a good dose of novelty and this gives value to the study. However, some things must be modified and others completed, especially at the level of microbial methodology. It details:
- In Abstract and text: "Pseudomonas", in italics. It is a microbial genus. Throughout the text, instead of using "Pseudomonas", you should use "Pseudomonas spp." since it is not known how many species and strains of the genus are detected in the counts. I do not detect a genus, in the abstract, but rather I detect all the species and strains of that genus that may be present.
- 2.1: 72 strains were isolated. From how many samples? How were they isolated? How was it determined that they were LAB?
- Table 2: the units of the growth response need to be indicated in the title. What do the numbers that appear in the table indicate?
- 3.2: How was the genus and species of the NML21 strain identified?
- 3.3: Were the 6 strains also genetically identified?
- Figs 3-7: as in Fig. 2, the titles must indicate what CK, M1, M5 and M9 means. Remember that a figure must be able to be interpreted without having to resort to the text
- Figs 2-7: use "treatments" in the titles since 4 treatments were applied
- Figs. 3, 6....; "A.bisporus" in italics
- line 359: "As expected, the strain NML21 converted..."
- the genus and species of the microorganisms must always be written in italics. Review all text, especially Discussion
- line 381: what is the identification of the strains used in the cited study. Only genus and species are indicated. Not all strains of that genus and species give the same result. Same line 396
- line 396: (Rhodotorula yeast Cryptococcus laurentii) ??
- line 427: "...acid, and a browning resistant..."
- line 428: "...from 72 strains for the..."
​
Round 2
Reviewer 2 Report
Comments and Suggestions for Authors
The authors have responded to the observations made, in general. There are only a few things left to correct, which are detailed:
- Table 3: use "treatments" in the title. Indicate in the Table that MN3....HG23 are strains
- lines 258 and 306: use "treatments" in titles
- line 417: Lactococcus lactis
​
​
Author Response
1.- Table 3: use "treatments" in the title. Indicate in the Table that MN3....HG23 are strains
Response to comment: We have revised "treament" to "treaments" in the title, and have revised the strain names in Table 3, as explained in the table notes.
2.- lines 258 and 306: use "treatments" in titles
Response to comment: We have revised "treament" to "treaments" in the title.
3.- line 417: Lactococcus lactis
Response to comment: We have revised it to "Lactococcus lactis A-NZ9000 (pVE3655)".